# Use of Membrane Technologies in Dairy Industry: An Overview

**DOI:** 10.3390/foods10112768

**Published:** 2021-11-11

**Authors:** Mònica Reig, Xanel Vecino, José Luis Cortina

**Affiliations:** 1Barcelona Research Center for Multiscale Science and Engineering, Campus Diagonal-Besòs, 08930 Barcelona, Spain; xanel.vecino@upc.edu (X.V.); jose.luis.cortina@upc.edu (J.L.C.); 2Chemical Engineering Department, Escola d’Enginyeria de Barcelona Est (EEBE), Campus Diagonal-Besòs, Universitat Politècnica de Catalunya (UPC)-BarcelonaTECH, C/Eduard Maristany 10-14, 08930 Barcelona, Spain; 3CETaqua, Carretera d’Esplugues, 75, 08940 Cornellà de Llobregat, Spain

**Keywords:** microfiltration, ultrafiltration, nanofiltration, reverse osmosis, resource recovery

## Abstract

The use of treatments of segregated process streams as a water source, as well as technical fluid reuse as a source of value-added recovery products, is an emerging direction of resource recovery in several applications. Apart from the desired final product obtained in agro-food industries, one of the challenges is the recovery or separation of intermediate and/or secondary metabolites with high-added-value compounds (e.g., whey protein). In this way, processes based on membranes, such as microfiltration (MF), ultrafiltration (UF), nanofiltration (NF) and reverse osmosis (RO), could be integrated to treat these agro-industrial streams, such as milk and cheese whey. Therefore, the industrial application of membrane technologies in some processing stages could be a solution, replacing traditional processes or adding them into existing treatments. Therefore, greater efficiency, yield enhancement, energy or capital expenditure reduction or even an increase in sustainability by producing less waste, as well as by-product recovery and valorization opportunities, could be possible, in line with industrial symbiosis and circular economy principles. The maturity of membrane technologies in the dairy industry was analyzed for the possible integration options of membrane processes in their filtration treatment. The reported studies and developments showed a wide window of possible applications for membrane technologies in dairy industry treatments. Therefore, the integration of membrane processes into traditional processing schemes is presented in this work. Overall, it could be highlighted that membrane providers and agro-industries will continue with a gradual implementation of membrane technology integration in the production processes, referring to the progress reported on both the scientific literature and industrial solutions commercialized.

## 1. Introduction

Dairy has been selected as one important agro-food industry and is a well-known producer in the European Union (EU), especially in Spain [1,2]. The EU’s dairy sector is the second biggest agricultural sector [3,4], producing 172.2 million tonnes of raw milk on farms in 2018 [5]. In 2018, most of the total raw milk produced was delivered to dairies for further processing (160 million tonnes), while the rest (12.2 million tonnes) was used on farms (either consumed, processed, directly marketed or used as feed) [5].

The milk delivered to dairies is processed into (i) fresh products, such as drinking milk, and other fresh products, such as yoghurts, cream and fermented milks, and (ii) manufactured products: cheese, milk powder, butter and whey, among others. Additionally, it is worth noting that butter and cream production is a process that generates skimmed milk, whereas the production of cheese, drinking milk and powdered milk is a process that consumes skimmed milk. Therefore, skimmed milk is a by-product generated during the production of butter and cream, which is used for the processing of other dairy products (e.g., powdered milk) [4,5].

One of the key processing tools in agro-food industries for the treatment of food products, as well as by-products or agro-food waste, is membrane technology [6,7,8,9,10]. In addition, the global market of membranes for food and beverage processing is estimated to achieve around USD 8.26 billion by 2024, an increase of 6.8% of the compound annual growth rate (CAGR) over the forecast period (2019–2024) [11]. Among them, the pressure-driven membrane processes, such as micro- (MF), ultra- (UF) and nano- (NF) filtration and reverse osmosis (RO), have been applied in agro-food industries to treat raw material streams and by-products [12,13,14,15,16,17]. The driving force of these membrane processes is the transmembrane pressure (TMP). Additionally, the key component in the membrane separation processes is the molecular weight cut-off (MWCO, usually expressed in Da) [12]. For that, pressure-driven membrane processes can be characterized according to these two parameters (TMP and MWCO) [13]. In this sense, MF requires > 100,000 Da and 0.1–2 bar; UF utilizes 1000–100,000 Da and 2–10 bar; NF uses 100–1000 Da and 5–40 bar; and RO needs 1–100 Da and 30–100 bar [18,19].

The largest share of the membrane market is for UF systems, accounting for 35% of the market, followed by MF processes (33%) and, lastly, NF/RO systems (30%). Otherwise, other filtration systems and membrane processes, such as electrodialysis (ED), pervaporation (PV) and membrane liquid contactors (MLCs), have only a small market share [20].

Compared to conventional methods, membrane technologies offer several advantages, including operation at a low temperature, the absence of phase transition, high separation efficiency, high productivity in terms of permeate fluxes, low energy consumption, simple equipment, simple operation and easy scale-up [12,20,21]. However, the bottleneck of membrane filtration is membrane fouling and concentration polarization phenomena, which cause a reduction in the flux and, consequently, process productivity losses over time [22]. The use of regular cleaning steps can minimize these phenomena [20].

Regarding the dairy industry, MF can be used as a pretreatment to remove both bacteria and fat, as well as to fractionate milk products. UF can be applied as a standardization process of milk; however, the breakthrough use of UF was to convert milk whey into refined proteins for commercial use. ED, with bipolar membranes (named EDBM), can be used to alkalinize acid whey [23]. NF can be utilized for whey demineralization [24] and RO for concentration steps.

Based on the above facts, research into the use of membrane-based processes in agro-food industries, in particular the dairy industry, is an interesting area for current membrane science and technology fields. Additionally, in view of the aforementioned importance of this sector, the main purpose of this manuscript is to assess the aspects of membrane usage feasibility in agro-food industries with a focus on (i) process intensification; (ii) environmental protection, (iii) waste minimization and (iv) added-value product recovery from some selected parts of the traditional processes of these industries.

## 2. Commercial Membranes in Dairy Industry

Some companies have been developing membranes for the dairy industry, such as MEGA, Novasep and GEA [25,26,27,28]. For instance, GEA proposed MF for bacteria reduction, milk protein fractionation, fat removal and lactose reduction; UF for protein concentration, protein standardization, high-grade lactose by decalcification (calcium removal) and yield increase; NF for concentration, partial demineralization, lactose reduction and acid and caustic recovery; and RO for pre-concentration, concentration and water recovery.

Some manufacturers, such as GE Osmonics, Dow Filmtec and Koch Membrane Systems, developed polymeric membranes (MF, UF, NF and RO) for dairy use, such as the processing of raw milk into milk products. Most of them were made by polyethersulfone (PES), polyamide (PA) and polyvinylidene fluoride (PVDF) among other polymers, such as poly(piperazine-amide) (PPZ) and polysulfone (PS), with different MWCOs, operational pH ranges and retention values (see Table 1).

For example, Cuartas-Uribe et al. [29] studied the separation of lactose from whey ultrafiltration permeate using nanofiltration. For that, in all NF experiments, the membrane used was DS-5 DL from GE Osmonics, with the TMP ranging between 0.5 and 2.5 MPa. Hinkova et al. [30] evaluated two different commercial nanofiltration spiral-wound membranes, NTR-7450-S2F (Nitto Denko) and FILMTEC NF270-2540 (Dow Filmtec), under various conditions for cheese whey fractionation and separation. Both membranes showed comparable ion rejections, whereas lactose apparent rejections on NTR-7450-S2F were in the range of 82–98% and slightly lower (82–90%) on the FILMTEC NF270-2540 membrane. The large consumption of water in the dairy industry makes water reuse a challenge; for that, Riera et al. [31] characterized and nanofiltered flash cooler condensates from direct ultra-high-temperature (UHT) treatments. The nanofiltration treatment was carried out in a pilot plant with a SelRO MPS-34 2540 B2X membrane (Koch Membrane Systems).

Polymeric membranes are more widely deployed due to their lower cost and lower energy requirements compared to ceramic ones. In contrast, ceramic membranes are more resistant due to their physical, hydrothermal, and chemical stability conditions (e.g., operational pH range from 0 to 14 and cleaning temperature up to 150 °C). Indeed, depending on the operational conditions or specific application, different materials (e.g., Si, Zr, Ti and Al oxides, which have different surface charge in solution), apart from silicon carbide (SiC), are used. Ceramic membranes are usually used for MF, UF and NF applications [32,33,34]. Moreover, ceramic membranes can be easily cleaned by NaOH, NaOCl, HNO_3_ and H_2_O_2_, which are standard sanitizing agents in the food industry, so one of its advantages is the capacity to be reused [35].

Recently, Samaei et al. [33] reviewed the most widely applied ceramic membrane manufacturers for MF, UF and MF applications, reporting their chemical composition, configuration and dimensions (see Table 1).

Table 1 shows that companies have tried to improve membrane area and packing densities and overcome the pressure drop problems of commercial ceramic membranes. For instance, TAMI industries introduced a new tubular configuration membrane, called IsofluxTM, for MF purposes, with a different number of channels to achieve a membrane filtration area ranging from 0.2 to 0.5 m^2^. Moreover, the Pall Corporation company also developed MF and UF membranes, with an asymmetric, tubular, and hexagonal geometry, named Membralox^®^, with multiple channels. In this case, MF membranes are alumina based, and their pore size ranges from 0.1 to 1.4 μm, whereas UF membranes are zirconia based with a lower pore size ranging from 20 to 100 nm. For this reason, they can increase the packing densities up to 240 m^2^/m^3^. Last, but not least, the Veolia Water Technologies company mechanically modified the conducts of the permeate to develop a new product (CeraMem^®^), which is able to overcome the pressure drop problem.

## 3. Market role of Membrane Technologies in Dairy Industry

Milk products have high nutritive value, and, for this reason, they are more regularly included in daily dietary routines. This fact is driving the market size.

A change in demographic shifts and dietary patterns (e.g., taste, health needs and/or nutritive value) has led to a high consumption of yogurt, cheese and cream, boosting product demand and acting as a major driving factor for market growth. Shifting trends toward balancing nutrition with flavor and a variety of products may act as a major contributing factor toward industry growth. Therefore, the continuously increasing consumption and production of dairy products coupled with a rise in the deployment of production automation processes have also been identified to boost market demand. Indeed, technology innovation, an increase in the production of dairy products in various countries (such as the Netherlands, Germany, India, and Australia) and the rising consumption of processed milk products have been identified to boost market expectations over the forecast period. Moreover, automation in dairy processing favors the industry’s growth since it reduces contamination and cost while increasing efficiency, meeting the required quality standards [36,37].

In fact, the global dairy processing equipment market size was around EUR 7285 million in 2016 [36], and the global world milk and milk product market in 2017 was around 835 million tons [37]. Moreover, in 2016, the USA market was valued at EUR 650 million, and it is predicted to grow at a CAGR of 6.1% (from 2017 to 2025). Therefore, it seems that processed milk product consumption is increasing while its production is decreasing, which is due to the reduction in the number of cows. Therefore, one of the major changes has been the increase in milk consumption per capita.

Recently, Konarev et al. [38] analyzed different factors that affect the development of companies in the dairy industry. The study used tools of production functions to determine the total factor productivity in microeconomics, as well as to determine the impact of technological innovation on overall factor productivity and corporate growth. The results showed that the main factor of growth was capital expenditure, whereas the impact of labor, based on employers’ wage and social security costs, was relatively low. The study also concluded that membrane cell technologies stimulated corporate growth in the dairy industry.

When reviewing the dairy industry process streams, the major processing equipment used comprises pasteurizers, membrane filtration apparatus, homogenizers, separators, evaporators and dryers and mixing and blending devices [36].

According to the dairy market distribution by equipment type [36], membrane filtration equipment (MF, UF, NF and RO) is widely used for processing milk and for other purposes in dairy industries. Indeed, membrane filters’ market share can be compared with other main processing technologies, such as mixing and blending, with values close to 10% of the total value.

Moreover, as mentioned throughout the manuscript, membrane filtration techniques can be used for several purposes, such as processed milk concentration (e.g., before transportation) or protein concentration, solid particle separation, fat fractioning, spore and bacteria removal and lactose recovery. For instance, it is possible to reduce the carbohydrate content in milk, as well as remove bacteria during the process of production.

Changes in national and international regulations, industry standards and consumer demand, and the replacement of outdated devices are some of the major driving factors for the market of membrane equipment. In this direction, specialized processes, including UF, which facilitate the recovery of milk solids, and the effective utilization of NF and RO systems resulted in processing effectiveness and the achievement of low-fat, low-sugar, cholesterol-free products. Nevertheless, it is expected that a decline in the price of milk products will have a negative impact on the market. Both membrane filtration and mixing and blending equipment are used to improve product quality by adding vitamins, calcium and other types of components. Then, a potential higher role in market perspectives is expected. For instance, Europe is considered a potential region for the segment due to a rise in the demand for healthier, palatable and protein-rich products [36].

The UF technique is one of the most used membrane filtration techniques in the dairy industry for the retention of high molecular weight and suspended solids (such as milk proteins) and lactose content reduction among other applications, using a semi-permeable membrane. Its relevance as a function of the dairy sector can be shown in the global UF market by application [39]. The UF milk market has been increasing over time. In fact, due to the growth of using this membrane process for traditional milk processing, nowadays, the milk produced by UF has over 50% protein and 50% less sugar compared to regular milk. Moreover, the milk market is becoming a premium market due to consumer’s demand for processed cheese food, clean-label products, more nutritional products, etc. Furthermore, value-added dairy product innovation, new packaging and growing R&D activities will cause the UF milk market to rise even more in the future [39].

## 4. Bibliometric Evolution of Membrane Technologies in Dairy Industry

A bibliometric analysis of scientific publications, based on the Scopus database, was performed to identify the global research trends regarding membrane technology applications in the dairy industry during the time from 1980 to 2021. This analysis could improve our understanding of the most popular membrane applications in this field and provide a comparison of the membrane market view. The distribution of annual publications and the evolution of the number of accumulated articles containing the words “dairy” and “membrane” in the title, abstract or keywords are depicted in Figure 1a.

As can be seen in Figure 1a, the trend increases over the years from 26 articles (1980) up to 218 articles, which are the maximum published articles in a year (accomplished during 2019 and also during 2020). This trend confirms that membrane technologies are on the rise in dairy industries, and the integration of them into conventional processes could be a sustainable option for dairy processing, following a circular economy scheme for product valorization and waste minimization.

On the other hand, Figure 1b shows the number of publications containing the same words (“dairy” and “membrane”) organized per country. As summarized in Figure 1b, the analysis demonstrated that the United States was the most productive country, with 706 documents regarding “dairy” and “membranes”, which implies a percentage of 17.9%. This leading country was followed by China (268 documents) and France (247 documents). Canada was the fourth country in the ranking (201 documents). Another two European countries appeared among the top 10 countries: Germany (153 documents) and Italy (132 documents). In the 10th position appeared Spain, with 124 publications on this topic.

Finally, Figure 1c was plotted to understand the evolution of the number of publications considering several membrane technologies, such as MF, UF, NF, RO and diafiltration, in dairy industries from 1970 to 2021. As reported in Figure 1c, diafiltration is the membrane technology with the lowest number of publications in the dairy field (maximum of 9 articles in 2018), followed by NF (a maximum of 18 publications during 2018) and RO (2015 was the year with the most publications, up to 21). MF and UF are the most researched membrane technologies in relation to the dairy industry: a maximum of 36 publications were published in 2020 for MF and dairy, while 42 articles on UF and dairy were published during 2018. Therefore, the research on membrane technology, especially on the most popular techniques, such as MF and UF, must be considered as a very relevant issue within the global investigation regarding chemical engineering processes in the dairy industry.

Therefore, research investment in the dairy industry is in concordance with the membrane technology market in this industry, aforementioned in Section 3. For instance, it is possible to conclude that UF is the most widely used process in the dairy industry as well as the most researched technology (with the highest number of publications). On the other hand, it can also be seen that the USA is the largest producer in the dairy industry, as well as the country with the highest scientific productivity.

## 5. Membrane Technology Processes in Dairy Industry

Membrane technologies can be used in the dairy industry for many applications, such as milk clarification or fractioning and a concentration increase in specific components or the separation of them, since they cover a huge range of pore sizes (from 0 to 2 μm) and MWCOs (from 1 to 100,000 Da). For instance, MF can be used for fat globule (10 μm) fractionation as well as bacteria and spore (1 μm) removal. UF can be used for casein micelles (100 nm) or serum protein (10 nm) separation, whereas NF and RO can be used for lactose (1 nm), salt (0.1 nm) and water recovery [41,42,43,44].

### 5.1. Milk and Cheese Production Process

Milk is basically an emulsion of fats in water, accompanied by dissolved and suspended compounds, such as proteins, lactose, mineral substances and organic acids, among others [45,46]. An average composition of whole cow milk is summarized in Table 2.

The composition of the fatty acids of whole milk is different regarding the degree of saturation and chain length. Its specific flavor and mouthfeel is given by the milk composition. Moreover, different texture characteristics between different dairy products could be due to the fat globule size. The desired texture and the mouth feeling of the final product are obtained using the adequate fat fraction [48]. However, because of a huge amount of fats, their separation from the liquid phase is the first process of milk production (see Figure 2a). Indeed, its content depends on the final milk product: (i) whole milk usually has between 3.5% and 3.9% fats, (ii) semi-skimmed milk contains fats in the range of 1.5–1.8%, and (iii) skimmed milk is composed of less than 0.3% of fats [49]. This fat control process is called the standardization step.

In the traditional method, milk fats are separated from the liquid phase by gravitational forces (Figure 2a). However, it is a slow and inefficient technique for milk production and is accompanied by a potential risk of food safety. Therefore, nowadays, the transport of the liquid phase to the outer edge of the separator is achieved by applying a centrifugal force [50,51].

Microorganisms can grow in the milk medium; for that reason, after the standardization process, skimmed milk should be treated by pasteurization or UHT processes to ensure microbiological clearness and assure the required quality for consumers. The heat process that destroys the pathogenic microorganisms in milk (e.g., 62.8 °C for 30 min) is named pasteurization, and when the milk is heated to even higher temperatures (over 140–150 °C) but just for two seconds, it is considered a UHT process [52]. Nowadays, both methods are used worldwide, and their use depends on the consumption habits and production conditions of each country [53].

Homogenization is the last stage before bottling the milk. This process consists of avoiding the formation of milk fat layers by reducing the fat globule size [54].

On the other hand, the coagulation of milk casein produces cheese. As observed in Figure 2b, the first step of cheese production is the standardization of the raw milk, which is the same as for milk production [55]. Coagulation or curdling is the next stage of cheese production. In this stage, milk is separated into solid curds and liquid whey, usually carried out by acidifying the milk and adding the enzyme rennet [56]. In the acidifying step, some acids can be used, although the most used starter is bacteria, such as *Lactobacillus* spp., lactococci, *Streptococcus thermophilus* and leuconostocs, which ferment milk sugars into lactic acid [57].

After the coagulation process, the liquid whey is separated from the solids via a draining stage. At this point, there are fresh cheeses that are completed, and other cheeses, namely, the hard ones, are heated, forcing the extraction of more whey [58].

Although traditional production schemes (Figure 2) for milk and cheese production do not include membrane processes, membrane technologies allow the separation of different milk components, such as fat globules, casein, lactose and bacteria. As reported by Brans et al. [45], MF, UF, NF and RO enable the optimization of dairy industry processes by recovering or fractionating components with special interests as food supplements, such as whey protein. Moreover, Lauzin et al. [59] compared the composition, rennet-induced coagulation kinetics and cheesemaking properties of UF, NF and RO concentrates. The results shown that the RO and NF milks impaired cheesemaking properties, which may be due to their higher salt content.

#### 5.1.1. Milk Fat Fractionation by Microfiltration

As previously mentioned, milk is mainly composed of fat, which is present in spherical globules with a diameter between 0.1 and 15 μm. Usually, the diameter of small globules is less than 2 μm, whereas globules with a diameter higher than 2 μm are considered large globules. Indeed, milk is mainly composed of small globules, since 80% or more of the total globules are represented by a diameter less than 1 μm. However, they represent a small fraction of the total milk fat volume, since 90% or more of the total volume are globules with a diameter ranging from 1 to 8 μm. Thus, MF technology allows the fractionation of milk fats due to its pore size [60].

Goudédranche et al. [54] performed the fractionation of small (<2 μm) and large (>2 μm) fat globules from whole milk and creams by a patented process using microfiltration technology. In this case, whole milk, at 50 °C and with a fat content from 3.9% to 12%, was treated by an MF ceramic membrane with a 2 μm average pore size diameter. In the first case, 700 L/h·m^2^ of permeate flux was obtained with 1.7% of fat content and a retentate stream containing 20% fat content. When treating the second batch of whole milk (with a higher percentage of fat content), a smaller permeate flux was obtained (250 L/h·m^2^), containing 6.9% fat content and a more concentrated retentate (30% fat content). Once small and large fat globules were separated, different mixtures could be used to produce different products, such as fresh cheese, Camembert and mini sweet cheese.

Kowalik-Klimczak [32] also proposed the integration of this technology for milk fat separation during the manufacturing of products with specific nutritional purposes (see Figure 3).

To summarize, authors reported that MF allowed the possibility to adjust the texture of dairy products, such as cheese. Moreover, the use of small globule fat fractions, obtained as MF permeates, led to more unctuous and finer textural dairy products compared to products made from untreated or large fat creams [60,61].

#### 5.1.2. Bacteria and Spore Removal by Microfiltration

As described in Figure 2a, after standardization, UHT and pasteurization processes must be carried out on the skimmed milk before homogenization. UHT is a more effective treatment than the pasteurization process; nevertheless, UHT can be more damaging to milk properties, since a higher temperature is applied [62]. In this sense, MF is an alternative to UHT for the removal of bacteria and spores from milk, since the chemical and organoleptic properties of the milk are not altered [20,63]. For instance, Kowalik-Klimczak [32] proposed an integrated scheme for milk processing by membranes. In fact, MF was suggested as the main technique used for bacteria removal in the production of extended shelf life milk (see Figure 4). The proposed MF was positioned after fat separation in order to remove particles with a diameter between 0.1 and 10 μm, such as fat, bacteria and spores from milk and whey.

In fact, the first commercial system was patented by Alfa-Laval Food & Dairy Engineering AB [64]. In the proposed process, raw milk is fractioned into cream and skim milk by a centrifugal separator. Subsequently, the skim milk is treated at a constant TMP by a microfiltration ceramic membrane (1.4 μm pore size), obtaining the separation of fat globules and bacteria. A skim milk with low bacterial content (<0.5% of the original value) is achieved in the MF permeate, whereas a high-fat content, mainly bacteria and spores, is obtained in the concentrate stream. Afterward, the retentate is mixed with the desired quantity of cream, after its standardization process, and submitted to a conventional UHT treatment: 130 °C for 4 s. The obtained product is mixed with the MF permeate and pasteurized. In this process, less than 10% of the raw milk is treated by UHT, i.e., not heated at a high temperature. Therefore, it is possible to significantly improve the milk sensory quality [20,64]. It is important to note that the Alfa-Laval company is a leading global provider of first-rate products in the areas of heat transfer, separation and fluid handling in many industries, including the dairy sector [65].

Furthermore, other authors have proposed MF as an efficient technique for bacteria and spore removal [60,66,67]. For instance, the use of MF ceramic membranes, with a pore size of 1.4 μm, operated at a constant TMP of 0.5 bar and a cross-flow velocity of 2.7 m/s was described by Saboya and Maubois [60]. The results showed a flux of 1.4 × 10^−4^ m/s and a reduction factor of bacteria and spores above 3.5. Furthermore, Guerra et al. [66] achieved the same flux at a cross-flow velocity of 1 m/s with a reversed asymmetric MF membrane with a pore size of 0.87 μm. The reduction factor of the bacteria and spores was between four and five.

Therefore, MF can be used for bacteria and spore removal to be able to produce other dairy products, since there are many applications in the dairy industry for bacteria-free milk, such as cheese production. In this case, low-bacteria milk improves cheese quality, removing its off-flavors. Moreover, in the production of whey protein concentrates and isolates, bacteria removal by MF increases the quality of the product and keeps the heat treatment to a minimum, which better preserves the functional properties of the whey proteins [20,68].

Although MF is the most proposed and recommended membrane technique for bacteria and spore removal, UF can also be used for large bacteria removal [69] or even to treat cold whey streams (<20 °C), which improve microbiological specifications, such as spore-forming bacteria [70].

#### 5.1.3. Whey Protein Concentration and Fractionation by Membrane Technologies

As described in Figure 2b, after the curdling process for cheese production, a liquid fraction, named whey, is obtained. Whey composition depends not only on the milk properties but also on the technology used, with 5.5–6.5% dry matter. In this case, lactose represents around 70–80% of the dry matter in whey, followed by proteins (about 10%), minerals, nitrogen elements, acids, fats and water-soluble vitamins [30]. Although traditionally whey was considered useless for humans and used for animal feed, nowadays, it is considered a source of valuable proteins, widely used in the food industry and nutritional supplement production [71]. Recently, Wen-Qiong et al. [72] and Kelly [73] also produced an overview of the current use of membrane materials and membrane processing in cheese whey protein recovery. However, the main drawback of whey valorization is the presence of fat because it decreases its functional properties and leads to a shorter storage time. For that, once the whey is drained out of cheese vats, it is defatted.

Indeed, two whey protein products can be mainly obtained in the dairy industry: whey protein concentrate (WPC), which is a concentrated protein solution (80%), and whey protein isolate (WPI), which can be produced by the separation of lactose and sugar from the whey protein concentrate. Kowalik-Klimczak [32] proposed the integration of UF and MF technology for cheesemaking treatment in order to produce whey proteins. UF was carried out to make cheese, obtaining whey protein concentrates (WPCs) as retentate. Moreover, if the retentate solution from UF was treated by MF, it was possible to obtain whey protein isolates (WPIs) as permeate (see Figure 5).

On the other hand, Maubois [74] and Fauquant et al. [75] developed the most common processes for fat reduction from whey, using the ability of phospholipids to aggregate, by calcium binding, under moderate heat treatment (8 min at 50 °C). Then, as shown in Figure 6, defatted whey can be obtained by MF, with a pore size of 1.4 μm, to separate the resulting precipitation. Subsequently, skim defatted whey (MF permeate) is filtrated by RO to concentrate the protein content up to 18–27% and remove undesired components [6]. Therefore, a purified skim whey is obtained after the RO process.

The obtained RO retentate can be used to produce whey powder, WPC and WPI or to perform whey protein fractionation [41,76,77,78]. For that, the purified skim whey (RO concentrate) can be purified by an UF membrane followed by a diafiltration treatment (see Figure 7) [77]. After the UF process, WPC is obtained in the retentate solution, with more than 77% of protein content. Then, WPI can be produced in the retentate stream of the diafiltration process, achieving more than 90% protein content. In this stage, both permeate streams (nanofiltrated and diafiltrated) are also valuable products due to their high lactose concentration [6,20].

On the other hand, if the retentate purified whey, after RO (see Figure 6), is vacuum evaporated and then spray dried, whey powder can be obtained. In this sense, Bédas et al. [79] proposed NF for the recovery of lactic acid from whey stream prior to evaporation and spray drying at a semi-industrial scale in order to obtain whey powder. For that, purified whey was treated by NF to concentrate and reduce its volume. After NF, it was possible to obtain a demineralized concentrated whey in the retentate stream before vacuum evaporation and spray drying. In that work, NF was able to selectively demineralize monovalent ions (50–60%) while keeping the divalent ion content constant. Regarding the physico-chemical properties, the dryability of the lactic acid whey concentrate was improved by the NF stage.

Furthermore, since cheese is produced by the coagulation of milk casein, it is interesting to increase casein content in cheese milk. Casein enrichment significantly improves rennet coagulability and optimizes the curdling process: curds are firmer and consequently lead to fewer fines in whey [6]. In fact, Daufin et al. [6] proposed to enrich milk casein content via an MF stage (see Figure 8). In that work, skim milk was circulated along an MF membrane with a pore size diameter of 0.2 μm (homogeneous Al_2_O_3_ membrane), obtaining a sweet whey-rich permeate and a retentate solution, enriched with a solution of native micellar calcium phosphocaseinate (NCPP) or also named as micellar casein concentrate (MCC). One option to treat this retentate solution is to purify it by diafiltration by dilution with water and to subsequently treat it by vacuum evaporation. The main advantage of the NCPP is that it has excellent rennet coagulation abilities. The coagulation time of the NCPP solution (3%) was reduced by 53% in comparison with raw milk. Additionally, the preparation of NCPP requires special membrane designs, such as UTP (uniform transmembrane pressure) cartridges and porosity gradient (GP^®^ or Isoflux^®^) membranes, to prevent flux decline during long-term operations.

Moreover, MF also reduces the detrimental effects of heat treatment on the rennet coagulability of milk due to the partial reduction of the whey proteins/caseins ratio. However, the MF permeate solution is also valuable since it can be processed to obtain WPC and WPI by UF, followed by diafiltration as showed in Figure 8 [6,60].

Moreover, whey protein concentration is also attractive since individual serum proteins can be isolated. In fact, as mentioned in Table 2, milk has some serum proteins, which are mainly β-Lactoglobulin and α-Lactalbumin, followed by proteose-pepton, immunoglobulins, bovine serum albumin (BSA) and lactoferrin. Indeed, some of these serum proteins have interesting physicochemical properties, such as β-Lactoglobulin, which can be used in emulsification, foaming and gelling, and α-Lactalbumin, for pharmaceutical applications. To summarize, these proteins can be obtained from defatted whey [45].

In fact, α-Lactalbumin polymerizes reversibly with residual lipids and other whey proteins except β-Lactoglobulin at a pH range between 4.0 and 4.5 and moderate heat treatment (30 min at 55 °C). Using an MF membrane, with a pore size of 0.2 μm, β-Lactoglobulin can be separated. Then, both permeate and retentate (β-Lactoglobulin and α-Lactalbumin) can be purified by solubilization and UF, using a 50 kDa membrane, as can be seen in Figure 9 [20]. Moreover, Heidebrecht et al. [80] tested different microfiltration membranes (ceramic standard and gradient), pore sizes (0.14–0.8 µm), transmembrane pressures (0.5–2.5 bar) and temperatures (10, 50 °C). The authors obtained inmunoglobuline G (IgG) transmission rates between 45% and 65%, while the casein fraction was reduced below 1% in the permeates, with a ceramic gradient membrane with a pore size of 0.14 µm. Toro-Sierra et al. [81] developed an integrated method with the potential to be scaled up and to be able to produce pure native fractions of α-Lactalbumin and/or β-Lactoglobulin, comprising as follows: (1) selective thermal precipitation of α-Lactalbumin, (2) aging of the formed particles, (3) separation of native β-Lactoglobulin from the precipitate via MF and UF processes, (4) purification of β-Lactoglobulin, (5) resolubilization of the precipitate and (6) purification of α-Lactalbumin. Afterward, it was possible to obtain α-Lactalbumin with a yield of about 60.7% and 80.4% and a purity of 91.3% and to produce β-Lactoglobulin with 97.2% purity and a yield between 80.2% and 97.3% as a function of the membrane operation parameters.

Gésan-Guiziou et al. [82] used polymerization and UF steps in order to purify both serum proteins, reporting 85–94% purity for β-Lactoglobulin and 52–83% purity for α-Lactalbumin.

#### 5.1.4. Lactose Recovery from Whey Processing by Membrane Technologies

As abovementioned, UF is used to obtain WPC from defatted whey. In this case, the permeate stream contains lactose, which is a valuable product in the dairy industry. For that, the permeate solution can be concentrated and recovered via an RO or NF stage [20]. For instance, Hinkova et al. [30] reported data for lactose desalination by UF and NF treatments from salty whey, following Figure 10.

In that work, a tubular ceramic UF membrane (MWCO 500 nm) provided by Membralox (Pall, New York, NY, USA) was used to treat whey by applying a constant transmembrane pressure of 2 bar. Then, the UF permeate was purified by NF at 60 bar and 900 L/h (maximum flow rate), using two different spiral-wound membranes, NTR-7450-S2F (Nitto Denko, Osaka, Japan) and FILMTEC NF270 (Dupont, Delaware, DE, USA), with polyamide as an active membrane layer.

The results showed minimum lactose losses during the UF process since lactose rejection was 1%. Additionally, the rejection of the largest protein in whey (BSA) was almost 100%. On the other hand, higher lactose rejections were obtained by NF, achieving the largest rejection value using the NTR-7450-S2F membrane (85–96%) in comparison with the membrane NF270 (81–88%). Moreover, by using the NTR-7450-S2F membrane, low monovalent ion (Na^+^, K^+^ and Cl^−^) rejections were obtained (between 5% and 16%) at pH values of 5.0–5.7, whereas around 50% of calcium was rejected. On the other hand, at the same pH, higher ion rejections were obtained using the NF270-2540 membrane: 7–26% for monovalent ions and >50% for calcium. Therefore, the membrane NTR-7450 is the most suitable for whey desalination (monovalent ion separation and around 50% calcium passage into the permeate), while recovering about 95% lactose in the concentrate.

On the other hand, from the same dairy by-product, e.g., whey, or from the recovered lactose, galactooligosaccharides (GOSs) can be obtained. Hence, during the hydrolysis of lactose, by means of the enzyme β-galactosidase, GOSs are produced. GOSs are considered as prebiotic substances widely used in different foodstuffs, such as baby foods, fruit juices, bakery products and candy [83]. In order to obtain a good quality of GOS, the nanofiltration process is generally applied [84,85,86,87,88,89,90], followed by an ultrafiltration membrane bioreactor (UF-MBR) [91,92,93]. For example, Michelon et al. [84] selected the NP030 polyethersulfone nanofiltration membrane for the purification of GOS from a solution mimicking transgalactosylation reactions catalyzed by b-galactosidase from *Kluyveromyces marxianus* CCT 7082. For the commercial mixture containing lactose, glucose and galactose, a high purification factor and permeate flux were obtained at 35 °C and 3 MPa, recovering 61% (*w*/*w*) of GOS.

#### 5.1.5. Whey Treatment by Nanofiltration

In fact, dairy and milk processing industries were some of the main promotors of the use of NF in the food industry, especially for whey protein valorization. This is mainly due to the MWCOs of NF membranes (100–1000 Da), whose values are between those of the UF and RO techniques. Indeed, NF has not been a standalone technology for dairy and milk processing, since it has been integrated with other membrane technologies in several steps of this industry, such as protein hydrolysate fractionation, concentration of whey protein, cheesemaking, effluent recovery and the purification of waste stream [24]. In fact, NF is postulated as an alternative to ED with capabilities for solute fractionation. NF membranes exhibit a low rejection for single-charged electrolytes (e.g., NaCl and KCl) but show very high rejection for multivalent electrolytes (e.g., MgCl_2_, Na_2_SO_4_ and MgSO_4_) and/or organic compounds (e.g., urea, lactose and proteins). The rejection behavior, although it is still not fully understood, is considered to be strongly dependent on dielectric exclusion for the case of charged species, solute properties and membrane properties [94,95]. Moreover, NF has some advantages in comparison with ED, such as operational cost reduction (electrical consumption and a reduction in wastewater disposal cost) and simultaneous demineralization and concentration of whey [24,96].

On the other hand, due to NF selectivity, it has been successfully introduced in the disposal of whey, which is one of the main problems of the dairy industry due to its high organic content. In that sense, organic acids, most of the single-charged ions and lactose content could pass through the NF membrane (permeate). Therefore, liquid whey has been partially demineralized and also concentrated by NF membranes [20].

Furthermore, NF is used for salt whey treatment, after the addition of NaCl to curd, and also for whey deacidification, by the addition of HCl to milk in casein production. In fact, it is necessary to demineralize whey for human and animal consumption. Then, after the NF process, high-quality products (e.g., whey powders), which are highly rich in proteins and nutrients, are produced. Moreover, NF is applied for scaling prevention and the removal of salts, and it is also used for lactose deashing. This membrane technology offers a significant improvement in Ca(II) transport with both processed and unprocessed whey and also better water recovery factors when integrated with acid whey permeate from a UF stage. To summarize, the integration of NF membranes, instead of evaporative stages and/or schemes, including ED, benefits the enrichment and demineralization of whey [24,97].

NF has also been used for simultaneous concentration and partial demineralization of cottage cheese whey by coupling it with continuous variable volume diafiltration. In that study, the demineralization extent of single-charged ions was satisfactorily high (more than 70%), and the retention of the useful components of whey, such as lactose and protein, was higher than 90% [98]. Moreover, the separation of lactate and lactose by integrating an NF stage could be achieved by increasing the pH above the dissociation constant of lactic acid with rejections above 90%, because as a weak acid, the dissociation state of lactic acid is pH dependent, following the Henderson–Hasselbalch equation [99]. On the other hand, NF has also been successfully applied for the concentration of tofu whey to produce two fermented lactic beverages with (a) 10% of concentrated tofu whey and 90% of milk and (b) with 20% of concentrated tofu whey and more than 80% of milk. The water recovery factors achieved were higher than 4.5. Additionally, the associated enhancement of the isoflavone content and antioxidant activity of the concentrated tofu whey improved its nutritional value [100].

Moreover, it is possible to obtain inhibitors of different bacteria by treating the UF permeate of whey protein tryptic hydrolysate by NF. Furthermore, the retentate stream, rich in peptides, from the tryptic hydrolysate of whey protein can be valorized as a natural bio-preservative [101].

To summarize, NF has become greatly valuable technology, since it permits partial demineralization and also a reduction in volume in a single step. For this reason, NF is used as an alternative to conventional processes for the separation of single-charged ions and also for the partial demineralization and concentration of whey. Finally, it is worth mentioning that NF linked with variable volume diafiltration or multistage batch diafiltration could also be used to enhance the demineralization degree of single-charged ions [97].

#### 5.1.6. Emerging Membrane Processes for Dairy Processing Effluents

There are other emerging membrane processes, such as forward osmosis (FO) and membrane distillation (MD), that must be mentioned for dairy stream treatment.

For instance, FO has been studied to produce whey powder from whey and water recovery [102] or to concentrate several dairy streams, such as whey protein solutions [103], skim milk and whey [104], or other relevant dairy streams (e.g., demineralized whey, sweet whey, WPC and lactose) [105]. Aydiner et al. [102] proposed the integration of an FO system previous to an RO when using NaCl as draw solution instead of a UF pretreatment. The results showed that the FO/RO integration resulted in higher water recovery percentages and better whey powder production. Wang et al. [103] used hollow fiber FO membranes obtaining high water fluxes, low reverse solute diffusion and high WPC retention (more than 99.9%) when using NaCl as draw solution. Moreover, they concluded that it could be possible to improve the water flux even more by increasing the draw solution concentration or the cross-flow velocity of the system or decreasing it via an increase in the WPC concentration in the feed solution. Chen et al. [104] proposed to use NaCl as draw solution, since it mimics the potential brine stream, which is available in dairy processing plants. The objective of the work was to concentrate skim milk and whey by FO. Indeed, it was possible to achieve concentration factors around 2.5 for both streams, which also resulted in a total solid concentration. The authors also reported that it was possible to increase the water flux, increasing the transmembrane pressure of the process. Additionally, they concluded that the FO technique required less specific energy than RO for milk and whey concentration. Finally, the system had some disadvantages, since it was not possible to reject small organic molecules (e.g., lactose) by using FO. The same authors used MgCl_2_ as draw solution to concentrate process streams, such as demineralized whey, sweet whey WPC and lactose, from a dairy industry using commercial FO membranes. The results indicated that it was possible to achieve high concentration factors (more than four for sweet whey), and it was possible to maintain the nutritional value of the treated streams (for instance, proteins and lactose were rejected by the FO system). Although intensive activity at the research level has been completed, the technology has not reached commercial scale. FO membrane producers have not reached large production capacity, and, still, two main challenges are waiting to be solved: (i) the regeneration of the solution, linked to the access to waste heat, which is not always available and (ii) the salt transport from the draw solution stream to the product solution stream.

Furthermore, another membrane technique of great interest in the agro-food industry, such as for dairy fluid processing, is MD. MD is an emerging thermal separation process based on vapor transport through a hydrophobic membrane due to the vapor pressure gradient. In fact, MD is presented as a competitive alternative to conventional methods, such as thermal evaporation [106,107], as it is a valid alternative in terms of energy consumption, final permeate quality and protein denaturation diminution [107,108]. However, its main drawback is the potential fouling and wetting of the membrane [108], such as calcium scaling [109], apart from the low evaporation fluxes in comparison with RO and thermal evaporation [107]. MD has been mainly studied to concentrate dairy products, such as milk and whey [107,110]. For instance, Gül et al. [110] used a hybrid system of osmotic distillation and MD to avoid brine dilution and to be able to concentrate milk at the same time as brine. The hybrid process resulted in higher fluxes in comparison to using only osmotic distillation. Moreover, Cassano et al. [107] were able to concentrate milk whey with high quality, minimizing protein denaturation, although concentration polarization phenomena occurred at high concentrations, diminishing the fluxes. On the other hand, Abdelkader et al. [108] studied several parameter effects when pretreating a saline dairy stream from a hard cheese industry, such as flow rate, temperature differences, feed content and organic load. The results indicated that a lower flow rate and a decrease in the temperature improved the membrane hydrophobicity, achieving lower fluxes. However, during all tests, membrane fouling occurred, and they concluded that the feed content and the organic load have a direct effect on the deposition layer. Kesia et al. [109] used MD as a technique to concentrate a by-product from the cheesemaking industry, which is produced from a salty whey. In that case, permeate flux was affected by protein presence in the feed solution, the cross-flow velocity and the membrane pore size. The results indicated that it was possible to concentrate the solids from the salty whey waste stream three times by MD.

Finally, Song et al. [111] proposed a hybrid system for dairy wastewater treatment, integrating both mentioned techniques: FO and MD. In that case, again, NaCl was used as draw solution in a cross-flow FO module and an air gap MD cell. The hybrid system was able to produce a high-quality permeate, obtaining high water and reverse draw solute fluxes, high contaminant rejection and high recovery rates, during long-term use.

### 5.2. Integration of Membrane Technologies in Dairy Industry Processing

Once different membrane processes have been revised to optimize and improve dairy industry operations, all of them would have been included in an integrated membrane technology process scheme for the dairy industry, considering milk and cheese production.

Figure 2 shows the traditional process for milk and cheese production, whereas the proposed combined scheme with membrane technologies is presented in Figure 11.

Comparing Figure 2 and Figure 11, it can be observed that several membrane processes have been proposed for milk and cheese production in dairy industries. Indeed, Figure 11 shows the proposal of the use of MF for milk fat fractionation (blue rectangles) before butter-making or fat standardization and also before milk homogenization or curdling cheese processes. MF was also planned for bacteria and spore removal before standardization/UHT and pasteurization techniques (green rectangle). Furthermore, whey protein concentration and fractionation have been considered by different membrane technologies. For instance, whey fat reduction and purification could be carried out via the integration of MF and RO techniques (orange rectangle) after curdling. Subsequently, from the purified skim whey, two options have been proposed: (i) UF followed by diafiltration for WPC and WPI production (purple rectangle) and (ii) NF, evaporation and spray drying for whey powder production (gray rectangle). Moreover, WPC and WPI production and casein (MCC) production and concentration were also planned from the low-bacteria skimmed milk permeate by the integration of MF, UF and diafiltration (black rectangle). Moreover, the fractionation of whey protein to obtain β-Lactoglobulin and α-Lactalbumin after polymerization has been proposed (maroon rectangle). Finally, lactose recovery from whey processing by UF and NF techniques has also been integrated in the proposed scheme (red rectangles).

To summarize, the integrated scheme with membrane technologies for milk and cheese production would improve the quality and texture of milk and cheese, and also, the whole process would be optimized, recovering added-value by-products, such as whey proteins, lactose and casein.

In this regard, Siebert et al. [112] showed the applicability of membrane filtration technologies as a tool able to create more functional food products. These potential new dairy products could be high-protein low-lactose fluid milk, high-protein low-lactose ice cream and non-fat yogurt made with fewer stabilizers. Moreover, this study concluded that (i) the added cost to produce functional food products is two to six percent of the existing retail price for similar standard dairy products as suggested by membrane manufacturing companies and that (ii) the most likely adopters of membrane technologies are yogurt manufacturers.

## 6. Membrane Fouling Mechanism in Dairy Processing

The area of priority for membrane manufacturers is to improve the permeate flux in order to attenuate membrane fouling. For that, there are some options available that can reduce membrane fouling (e.g., pretreat the feed stream, modify the module processes, modulate the characteristics of the membrane surface, compaction or measure the effective de-fouling) [113]. Pressure-driven membrane processes, including MF, UF, NF and RO, have been used to remove bacteria from whey, concentrate and demineralize whey, fractionate whey proteins, recover proteins from dairy industry process waters and other purposes. However, the required pumping energy and the high viscosity of dairy industry liquids limit the maximum attainable concentration factor. Moreover, irreversible fouling under high pressure usually renders membrane flux recovery difficult despite cleaning. For instance, sweet and acid whey treatment by MF or UF membranes causes long-term fouling and a progressive decrease in membrane lifetime [114]. Indeed, the observed flux and retention behavior in UF has been related to many fouling mechanisms (i.e., hydrophobic interactions, electrostatic interactions, solute adhesion, microbial fouling, particle size, ionic strength, membrane surface chemistry and protein properties (i.e., size and aggregation behavior)) [41,115,116].

In order to mitigate membrane fouling, pretreatment of the feed solution is usually the first choice, since it is a highly adaptable process. For instance, the pretreatment step can be adapted depending on the application, the membrane performance, the quality of the feed solution or the permeate requirements. Usually, media filters are used to pretreat the feed water before the membrane processes. However, these conventional pretreatments are not enough to meet the membrane requirements in the dairy process application. Additionally, other conventional treatments, such as biological and physicochemical treatments, biological treatment or a mixture of both processes (e.g., sedimentation, coagulation–flocculation, adsorption, conventional filtration and oxidation), are not applicable in most cases [42].

For membrane processes, the permeate flux declines by bio-fouling and scaling events [117,118]. Moreover, concentration polarization is inevitable and more severe in the operation of NF and RO, including processing trains. Moreover, an increase in specific electrical energy consumption due to the high cross-flow velocity that is required to control fouling events is also associated. For this reason, the main efforts of the principal membrane technology providers are to develop new strategies to control/mitigate fouling and to be able to avoid further complications in the equipment and operation. Furthermore, a list of other factors is necessary to reach the goal of high retention and/or permeation of target components in the streams to be processed.

The improvement in the hydrodynamic conditions on NF and RO membranes is used to control fouling. Thus, cross-flow velocity, shear stress, flow pattern at the membrane surface and feed flow rate are some of the main studied parameters for fouling mitigation. The improvement in these conditions could be an option to reduce concentration polarization on the membranes, since it could be possible to rise the mass transfer coefficient and turbulence. Furthermore, a new development to enhance membrane processes and mitigate the flux decline has been studied: air/gas sparging. In this case, air/gas bubbles are injected into the feed solution [119]. For instance, Patel et al. [119] observed an appreciable reduction in concentration polarization resistance in the presence of gas sparging. However, shear-enhanced filtration systems, such as a rotating disk membrane module, can also reduce the concentration polarization capacity and fouling since a very high shear rate at the membrane surface without a pressure drop is generated [120].

To summarize, de-fouling or integrating membrane systems with better cleaning performance is a high-priority research domain, and the development of new module designs and new materials could give interesting alternatives. For that, more research studies on fouling reduction, membrane lifetime increase and permeate flow rate optimization is needed for further development of membrane technologies.

## 7. Concluding Remarks

This manuscript covers an overview of membrane technology processes in an agro-food market industry: the dairy industry. Membrane technologies have demonstrated their potential to become a reality for separation and purification in milk and cheese processing industries, although some progress could already be made. Nowadays, a single separation technique could not be successfully used as a standalone system, but hybrid processes are characteristics of the commercialized and implemented solutions. For that, different membrane processes, such as MF and UF, and their integration have been proposed to be incorporated into conventional procedures of dairy industries to separate, produce and/or recover traditional products (such as milk) as well as new high-added-value products (such as whey, casein and GOS). The most widely used membrane process is MF, which can be used for milk fat fractionation, bacteria and spore removal, casein production, whey fat reduction or as a pretreatment for whey protein fractioning. However, UF has the largest market share. Accordingly, UF technology is also widely used in dairy treatments, combined with other membrane techniques, such as for lactose recovery as a pretreatment step of an NF procedure, as a pretreatment step of a diafiltration stage for WPC and WPI production from purified skim whey and for the fractionation of whey protein to obtain β-Lactoglobulin and α-Lactalbumin.

NF has been successfully implemented combined with other suitable membrane-driven techniques. The improvement in the design and fabrication of composite membranes is the main focus to satisfy the need of new applications. For that, several membrane properties are tuned: thermo-mechanical stability and physicochemical and morphology properties (e.g., zeta potential, hydrophilicity, charge density and porosity). Moreover, several functionalities are also modified, such as photosensitive, antimicrobial and adsorptive capabilities. In addition, process intensification is becoming a leading priority area of innovations as demonstrated by the solutions developed by providers such as Novasep, Koch Membrane Systems, GE_Osmonics and Pall Corporation.

However, RO is used for skim whey purification, after being removed by MF. Additionally, during the purification processes carried out by NF and RO, retentate streams are produced with high concentration values. Therefore, their treatment is an inconvenience for the implementation of these techniques. Moreover, membrane fouling on NF and RO is going to be a relevant area of research activity, but the sustainable management of the dairy industry contents constitutes a popular research area for this industry. Consequently, the most important areas for keeping the research on membrane technologies are the reuse, component valorization and retentate streams discharge. However, new initiatives to promote the recovery of additional valuable components from such streams are a topic on the research agenda of many industries and funding bodies worldwide. Finally, the evolution of membrane applications in this agro-food sector has demonstrated that membrane technologies have an increasing trend with respect to the dairy field, with MF and UF being the best filtration options. However, in recent years, the main innovations have been directed to other membrane techniques, such as NF/RO and FO/MD. For that reason, the successful integration of these technologies is expected, centered on (i) new NF membranes with new active-layer surface chemistries to exploit separation factors or promote a specific removal of un-desired ionic components; (ii) the integration of ED and ED with bipolar membranes to promote added-value by-product recovery, taking into account the development of more mono-selective membranes and bipolar membranes by companies such as Amstom, Suez and FujiFilm; and (iii) the promising development of FO membranes, such as those completed by the Aquaporin company. For such purposes, the future perspective will be the integration of NF/RO and ED with MF and UF as polishing steps.

## Figures and Tables

**Figure 1 foods-10-02768-f001:**
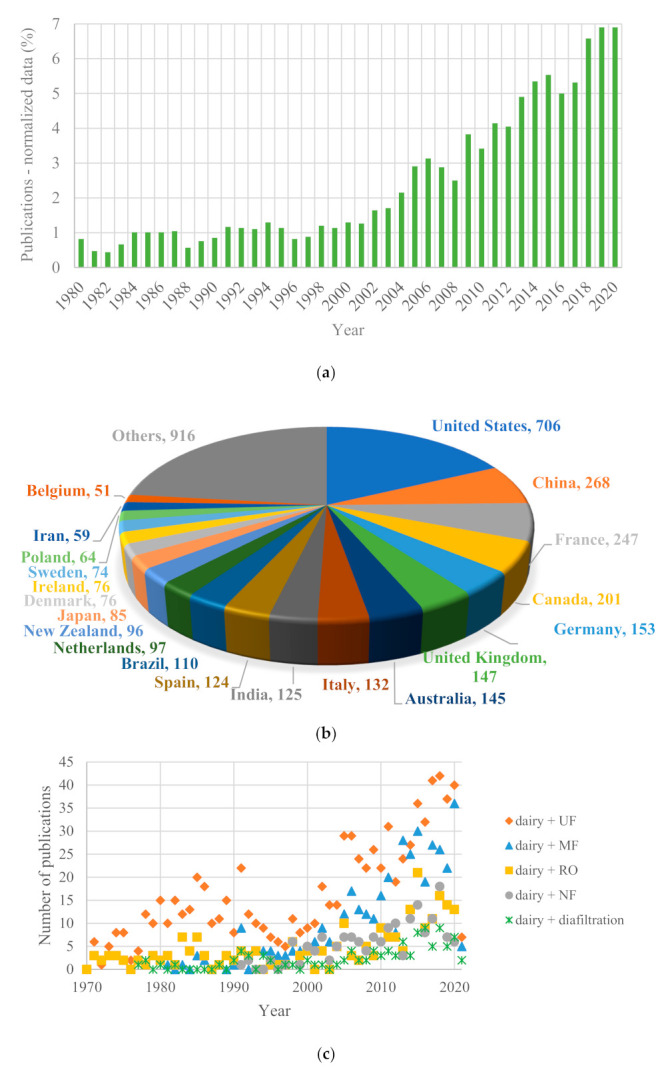
(**a**) Number of publications regarding “dairy” and “membrane” per year; (**b**) number of publications (from 1980 to 2021) regarding “dairy” and “membrane” per country; and (**c**) number of publications regarding “dairy” and “UF or MF or NF or RO or diafiltration” per year, from the Scopus database [40].

**Figure 2 foods-10-02768-f002:**
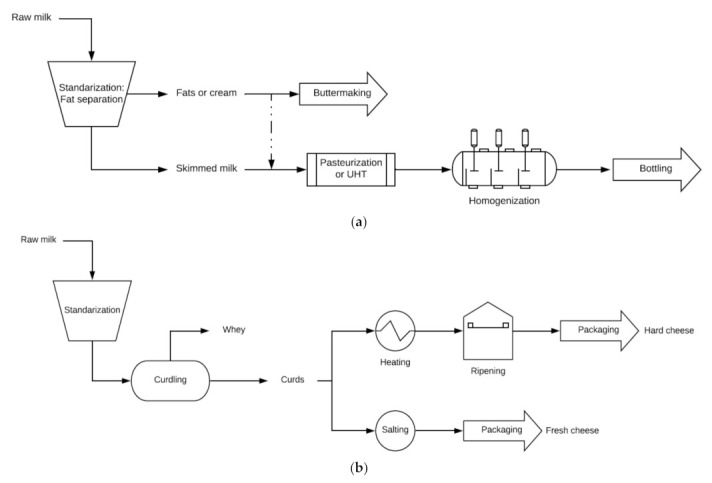
Traditional process schemes for (**a**) milk and (**b**) cheese production.

**Figure 3 foods-10-02768-f003:**
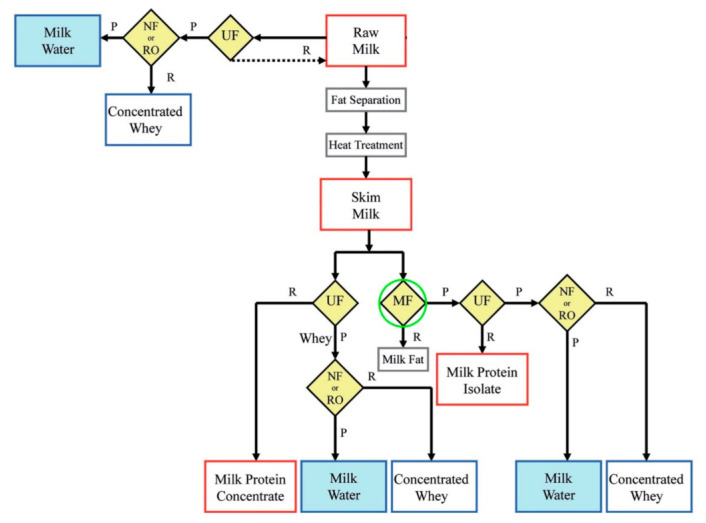
Process scheme for milk treatment and production of by-products with specific nutritional purposes (e.g., whey and milk protein), where MF is used for fat separation (adapted from [32]).

**Figure 4 foods-10-02768-f004:**
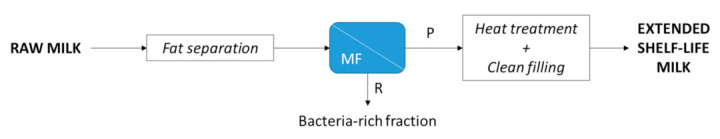
Process description for extended shelf life milk production (adapted from [32]).

**Figure 5 foods-10-02768-f005:**
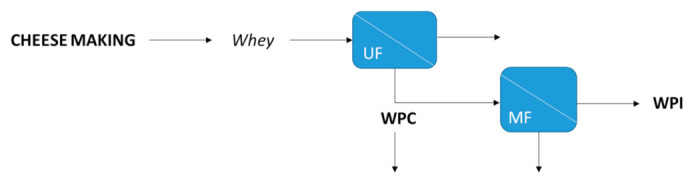
WPC and WPI production from cheesemaking by integration of UF and MF (adapted from [32]).

**Figure 6 foods-10-02768-f006:**
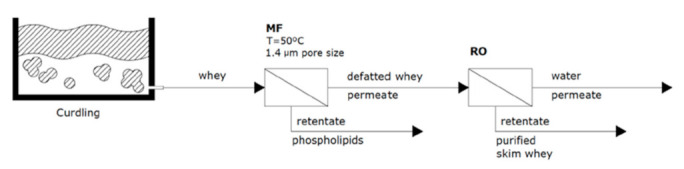
Whey fat reduction and purification by MF and RO membrane integration.

**Figure 7 foods-10-02768-f007:**
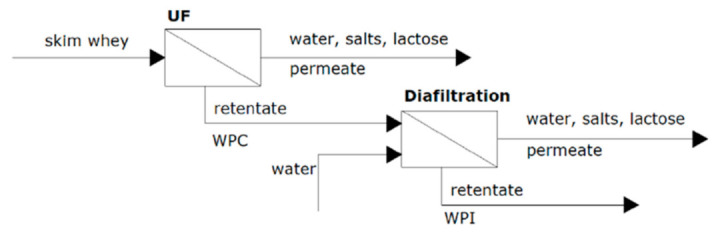
WPC and WPI production by UF and diafiltration from skim whey.

**Figure 8 foods-10-02768-f008:**
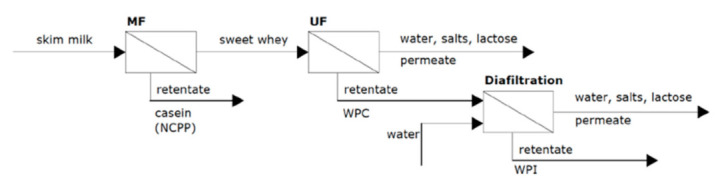
MF of skimmed milk to obtain NCPP and UF with diafiltration to obtain WPC and WPI from MF permeate.

**Figure 9 foods-10-02768-f009:**
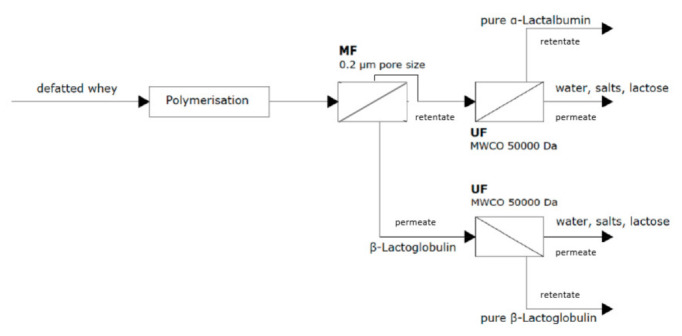
Membrane fractionation of whey protein to obtain β-Lactoglobulin and α-Lactalbumin by integration of MF and UF technologies.

**Figure 10 foods-10-02768-f010:**
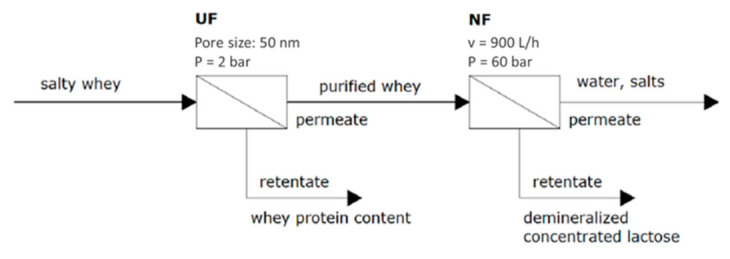
Demineralization of a high-lactose-content stream by integration of UF and MF.

**Figure 11 foods-10-02768-f011:**
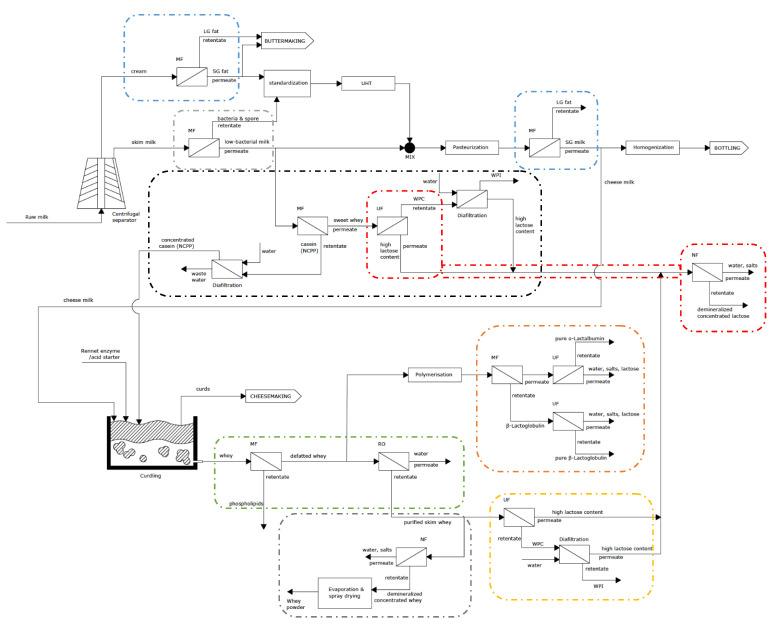
Membrane integration in dairy industries for milk and cheese production.

**Table 1 foods-10-02768-t001:** Overview of commercial membranes used in the dairy industry.

(a) Polymer-Based Pressure Membranes (Reproduced from [32], with Permission from Elsevier, 2021)									
Manufacturer		Symbol	Process	Polymer	MWCO/Pore Size	pH	Retention (%)	Temperature (°C)	
GE Osmonics		DL	NF	PPZ	150–300 Da	3–9, 2–10	98	50	
TriSep		TM10	MF	PVDF	0.2 µm	1–12	-	45	
		UF5	UF	PES	5000 Da	1–12	-	50	
		TS40	NF	PPZ	~200 Da	2–11	90, 40–60	45	
		TS80	NF	PA	~150 Da	2–11	99, 80–90	45	
		XN45	NF	PPZ	~500 Da	2–11	95, 10–30	45	
Synder		FR	MF	PVDF	800 kDa	3–9, 2–11	-	55, 50	
		V0.1	MF	PVDF	0.1 µm	3–9, 2–11	-	55, 50	
		V0.2	MF	PVDF	0.2 µm	3–9, 2–11	-	55, 50	
		BN	UF	PVDF	50 kDa	3–10, 2–11	-	60, 85	
		MK	UF	PES	30 kDa	3–9, 2–11	-	55, 50	
		ST	UF	PES	10 kDa	3–9, 2–11	-	55, 50	
		NFX	NF	PA	150–300 Da	3–10, 2–11	99, 40	50	
		NDX	NF	PA	~800–1000 Da	3–10.5	90, 30	50, 40	
		NFG	NF	PA	~600–800 Da	4–10	50, 10	50	
		NFW	NF	PA	300–500 Da	4–9, 3–10	97, 20, 98	50, 40	
Nanostone^TM^		PV650	MF	PVDF	0.31 µm	2–10, 2–11.5	-	60, 50	
		PE5	UF	PES	6 kDa	2–10	-	-	
		PE10HR	UF	PES	10 kDa	2–10	-	-	
Microdyn Nadir^TM^		P010	NF	PES	–	0–14	35–75	95	
		P030	NF	PES	–	0–14	80–95	95	
Dow Filmtec		NF	NF	PA	~200–400 Da	2–11	99	45	
Koch Membrane Systems		Dairy-Pro^TM^MF-0.1	MF	PES	0.1 µm	2–10, 2–11	-	50	
		Dairy-Pro^TM^UF-5K	UF	PES	5 kDa	2–10, 2–11	-	55, 50	
		Dairy-Pro^TM^UF-10K	UF	PES	10 kDa	2–10, 2–11	-	55, 50	
		Dairy-Pro^TM^MPS-34	NF	PSU	~200 Da	0–14	95	50	
		Dairy-Pro^TM^MPS-36	NF	-	1 kDa	1–13	10	50	
		Dairy-Pro^TM^NF-200	NF	PA	~200 Da	4–10, 2–11	-	50, 60	
		Dairy-Pro^TM^RO	RO	PA	-	4–10, 2–11	-	50, 60	
**(b) Ceramic Membranes (Reproduced from ([33], with Permission from the Author Kowalik-Klimckaz, 2021))**									
**Company**	**Product**	**Geometry**	**Designation**	**Membrane Material**	**Pore Size/MWCO**	**Available Length (s)-(mm)**	**Number of Channels**	**Outer Dia (mm)**	**Cannel Dia (mm)**
TAMI Industries	INSIDE CéRAM	tubular	MF UF Fine UF	–	–	580, 850, 1020, 1178	7, 8, 11, 19, 23, 25, 37, 39, 93	25, 41	1.6, 2.5, 3.5, 3.6, 4.6, 5.5, 6
	Filtanium^TM^	Tubular	MF UF Fine UF	–	–	580, 1178	8, 23, 39	25	2.5, 3.5, 6
	Isoflux^TM^	Tubular	MF	–	–	1020, 1178	8, 23, 39	25	2.5, 3.5, 6
	Eternium^TM^	Tubular	–	–	–	1178	7, 8, 23	25	3.5, 6
Atech Innovations GmbH	atec Ceramic membranes	Tubular	MF and UF	MF: α-Al_2_O_3_, TiO_2_, ZrO_2_ UF: TiO_2_, ZrO_2_, Al_2_O_3_	MF: 1.2, 0.8, 0.4, 0.2, 0.1 µm UF: 0.05 µm, 150, 100, 20, 10, 5, 1 kDa	1000, 1200, 1500	1, 7, 19, 37, 61, 85, 211	10, 25.4, 30, 41, 52, 54	2, 2.5, 3.3, 3.8, 4, 6, 8, 16
Pall Corporation	Pall^®^ Membralox^®^ IC	Tubular (Hexagonal)	MF and UF	MF: α-Al_2_O_3_ UF: ZrO_2_	MF: 0.8, 0.2 µm UF: 100, 50, 20 nm	1020	48	38, 43	4
Pall Corporation	Pall^®^ Membralox^®^	Tubular (Hexagonal)	MF and UF	MF: α-Al_2_O_3_ UF: ZrO_2_	MF: 1.4, 0.8, 0.5, 0.3, 0.1 µm UF: 100, 50, 20 nm	1020	19, 37	28, 31, 38, 43	3, 4, 6
Veolia Water Technologies	CeraMem^®^	Tubular	MF and UF	MF: mixed oxide, α-Al_2_O_3_, SiC, TiO_2_ UF: SiC, SiO_2_, TiO_2_	MF: 0.1, 0.2, 0.5 µm UF: 0.01, 0.005 µm, 50 nm	864	–	142	2, 5
ItN Nanovation AG	CFM Systems^®^	Flat sheet	MF	α-Al2O3	0.2 µm	L = 530 W = 6.5 H = 110	21	–	3
Meidensha Corporation	Ceramic flat sheet membrane system	Flat sheet	MF	α-Al2O3	0.1 µm	L = 1046 W = 12 H = 281	–	–	–
LiqTech International Inc.	CoMem^®^ Conduit	Tubular	–	SiC	–	865	–	146	3
LiqTech International Inc.	CoMem^®^	Tubular	–	SiC	–	305, 1016, 1178	–	25	3
Inopor^®^	Ceramic inopor^®^ membrane	Tubular	NF	Support layer: Al_2_O_3_ Membrane layers: TiO_2_ or SiO_2_	MWCO: 750, 600, 450 Da	1200	1, 4, 7, 19, 31	10, 20, 25, 41	3, 3.5, 6, 6.1, 7, 15.5
Cembrane	Cembrane Ceramic membrane	Flat sheet	MF	SiC	0.1 µm	L = 532 W = 11 H = 150	–	–	–

**Table 2 foods-10-02768-t002:** Typical composition of whole cow milk [45,47].

	Concentration (g/L)	Size Range
Solids-not-fat	8.9	-
Fat in dry matter	31.0	-
Fats	4.0	100–15,000 nm
Protein	3.3	-
Casein (in micelles)	2.6	20–300 nm
Serum proteins	0.7	3–6 nm
Lactose	4.6	350 Da
Organic acids	0.2	-
Mineral substances	0.7	-
Others	0.2	-
Water	871	-

## Data Availability

Data presented in this study are available on request from the corresponding author.

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
