# Peer review of "Use of Membrane Technologies in Dairy Industry: An Overview"

_foods, 2021, doi:10.3390/foods10112768_

Round 1
Reviewer 1 Report
This review paper presents a considerable amount of information on the use of membrane technologies in the dairy industry. It will provide a good reference for the dairy industry and dairy technologists.
There are some points which require the authors’ attention
- The title is unnecessarily long and convoluted. I’m still not certain what it means. The focus of the title seems to be on integration of membrane processes but that represents only a small part of the paper (section 5.2). I suggest the authors change the title to a shorter one which is more representative of the contents of the paper.
- In the first two pages there are several words which are incorrectly hyphenated. Please correct.
- Lines 47-48. Butter and “yellow products” are not processes as indicated. By the way, what are “yellow products”; I am not familiar with the term. Also, what is meant by “cream fabrications”?
- Lines 55-56. Please check the figure of 1.9% as it does not seem to match the data in lines 41-42 where 12.2 million tonnes out of 160 million tonnes were “used in other ways”
- Lines 60-61. Please indicate whether the figures are for membrane hardware or membrane-processed products
- Lines 305-313. Please eliminate repetition in this paragraph
- Line 316. “Limpidity” is an unusual word to use in this context.
- Lines 317-318. Pasteurization is designed to destroy pathogenic microorganisms; it does not destroy all microorganisms as the statement suggests.
- Line 320. What evidence is there for this statement? This is certainly not the case in many countries.
- Figure 5. Please adjust the MF symbol so that the diagonal runs in the same direction as in the other figures. Also indicate permeate and retentate
- Line 451 Change “whey proteins” to “whey protein products”
- Lines 455-458 and Figure 6. MF of WPC concentrate produces a phospholipid-enriched fat concentrate as the retentate and WPI solution as the permeate (see Figure 7). Please remove the term “milk water” as it is meaningless here; it should be reserved for product of RO/NF as in Figure 4. Please correct Figure 6.
- Line 505. The units of coagulation time cannot be percentage. Please correct.
- Figure 10 is incorrect. The retentate from MF is α-lactalbumin and the permeate contains β-lactoglobulin. The Gésan-Guiziou et al. reference in line 535 clearly shows the polymerized α-lactalbumin precipitates and hence would not pass through a membrane. Please correct this figure and also the relevant part of Figure 12.
- Line 595-596. Who says whey is not allowed for human consumption? Reference?
- Table 3 conveys very little information and should be deleted
- Line 799. Whey is not a “compound”. State what polysaccharides are referred to here.
- Line 816. This sentence is not correct. According to Fig 7, MF is used to remove fat from whey and RO is used to purify the permeate
Author Response
This review paper presents a considerable amount of information on the use of membrane technologies in the dairy industry. It will provide a good reference for the dairy industry and dairy technologists.
We acknowledge the positive judgment of the reviewer.
There are some points which require the authors’ attention
- The title is unnecessarily long and convoluted. I’m still not certain what it means. The focus of the title seems to be on integration of membrane processes but that represents only a small part of the paper (section 5.2). I suggest the authors change the title to a shorter one which is more representative of the contents of the paper.
Sorry for the inconvenience. The title has been changed according to reviewer suggestion.
- In the first two pages there are several words which are incorrectly hyphenated. Please correct.
Thank you for your remark. The words have been checked along the manuscript.
- Lines 47-48. Butter and “yellow products” are not processes as indicated. By the way, what are “yellow products”; I am not familiar with the term. Also, what is meant by “cream fabrications”?
The sentence has been corrected, since it was difficult to understand.
- Lines 55-56. Please check the figure of 1.9% as it does not seem to match the data in lines 41-42 where 12.2 million tonnes out of 160 million tonnes were “used in other ways”.
The information has been checked in the published literature and the numbers are correct, following citation nº5. 12.2 million tonnes are used on farms (processed or not), whereas the 1.9% is non-processed milk. Maybe the first sentence was not well written and it could produce a misunderstanding. The sentence has been changed.
Moreover, the last sentence has been removed to accomplish a suggestion of another reviewer.
- Lines 60-61. Please indicate whether the figures are for membrane hardware or membrane-processed products.
These values are for the global market of membranes. We understand that this means membrane hardware. The sentence has been modified in order to be more clear.
- Lines 305-313. Please eliminate repetition in this paragraph.
As suggested by the reviewer the paragraph has been modified to avoid repetition.
- Line 316. “Limpidity” is an unusual word to use in this context.
It has been changed to a synonym.
- Lines 317-318. Pasteurization is designed to destroy pathogenic microorganisms; it does not destroy all microorganisms as the statement suggests.
Thank you for your remark. The word "pathogenic" has been added for clarification.
- Line 320. What evidence is there for this statement? This is certainly not the case in many countries.
As indicated by the reviewer, the statement was not true for all countries. Nowadays, both methods are used worldwide and its use depends on consumption habits and production conditions of each country. For this reason, the sentence has been modified.
- Figure 5. Please adjust the MF symbol so that the diagonal runs in the same direction as in the other figures. Also indicate permeate and retentate.
The MF symbol and the diagonal runs have been modified in order to follow the same direction than in the order figures. Permeate (P) and retentate (R) are also indicated in the new version of the Figure (Figure 4 in the new version of the manuscript).
- Line 451 Change “whey proteins” to “whey protein products”.
Thank you. It has been changed.
- Lines 455-458 and Figure 6. MF of WPC concentrate produces a phospholipid-enriched fat concentrate as the retentate and WPI solution as the permeate (see Figure 7). Please remove the term “milk water” as it is meaningless here; it should be reserved for product of RO/NF as in Figure 4. Please correct Figure 6.
We appreciate the reviewer comments since there was a typo error in Figure 6 (Figure 5 in the new version of the manuscript). The error has been corrected, the text has been modified accordingly and also “milk water” has been removed from Figure 6 (Figure 5 in the new version of the manuscript). Now Figure 6 and Figure 7 (Figures 5 and 6 of the new version of the manuscript) are consistent between them.
- Line 505. The units of coagulation time cannot be percentage. Please correct.
Thank you. It has been corrected.
- Figure 10 is incorrect. The retentate from MF is α- lactalbumin and the permeate contains β-lactoglobulin. The Gésan-Guiziou et al. reference in line 535 clearly shows the polymerized α-lactalbumin precipitates and hence would not pass through a membrane. Please correct this figure and also the relevant part of Figure 12.
As mentioned by the reviewer, the MF retentate is α- lactalbumin whereas the permeate contains β-lactoglobulin. Figure 10 (Figure 9 of the new version of the manuscript) and the text has been modified accordingly. Moreover, the relevant part of Figure 12 (Figure 11 of the new version of the manuscript) has been also modified.
- Line 595-596. Who says whey is not allowed for human consumption? Reference?
Thank you for this remark. The sentence has been modified in order to be consistent with Nath et al. (2018) - Reference 24.
- Table 3 conveys very little information and should be deleted.
As suggested by the reviewer, Table 3 has been removed.
- Line 799. Whey is not a “compound”. State what polysaccharides are referred to here.
The sentence has been modified.
- Line 816. This sentence is not correct. According to Fig 7, MF is used to remove fat from whey and RO is used to purify the permeate.
The sentence has been modified as indicated by the reviewer. Now it is consistent with Figure 7 (Figure 6 of the new version of the manuscript).

Reviewer 2 Report
This manuscript reviews the use of filtration technologies in the dairy industry. I really appreciated the section on the market description, and on the description of the new membrane products available.
My major remark is that membrane processes are presented as a potential technology for the dairy sector (for example lines 27-28), but in my opinion, they are rather mature technologies that are well implemented. In fact, I did not see a lot of novelty when a description of all the possibilities to process dairy by-products were presented. The manuscript could be shortened to highlight its novelty.
Minor comments
Lines 18-19. The authors referred to highly-added value compounds, but whey is between brackets. Please specify whey protein.
Line 21. Cheese whey
Line 26. Trains ? (I think another word should be used)
Lines 46-47. I do not understand the idea of this sentence.
Line 52. Remove the hyphen in com-plete. (there are so much extra hyphen in the manuscript. Please do a thorough review)
Lines 66, 74, 90, 93, etc.. Remove the hyphen : Mem-brane
Line 108 : Incomplete abbreviation for pvdf
Line 109 : PS instead of PUS ?
Line 149. Superscript for m2. (same comment at line 154)
Rephrase line 159.
Table 1. Table should be presented in table format. Not image.
Table 1 b), Quality too low for reading.
Lines 799-801. I am not sure about this sentence. At the line 72, it is mention that UF has the largest market share.
Author Response
This manuscript reviews the use of filtration technologies in the dairy industry. I really appreciated the section on the market description, and on the description of the new membrane products available. My major remark is that membrane processes are presented as a potential technology for the dairy sector (for example lines 27-28), but in my opinion, they are rather mature technologies that are well implemented. In fact, I did not see a lot of novelty when a description of all the possibilities to process dairy by-products were presented. The manuscript could be shortened to highlight its novelty.
We acknowledge the comments of the reviewer on the manuscript. Thank you very much for the positive comments about the market and new membranes description sections. On the other hand, as suggested by the reviewer, sentence of lines 27-28 has been slightly modified in order to remark that membrane technologies in dairy industry are presented. On the other hand, although, the proposal of reducing the paper was facing a previous revision where it was suggested to add more information in order to have more quality as a review, the text has been changed and slightly shortened. Moreover, a table and two figures have been removed in the new version of the manuscript Finally, all suggestions and comments provided by the reviewer have been addressed in this letter and have been considered in the preparation of the revised manuscript.
Minor comments
Lines 18-19. The authors referred to highly-added value compounds, but whey is between brackets. Please specify whey protein.
Thank you for your remark.
Line 21. Cheese whey
It has been changed.
Line 26. Trains? (I think another word should be used).
It has been modified.
Lines 46-47. I do not understand the idea of this sentence.
The sentence has been modified.
Line 52. Remove the hyphen in complete. (There is so much extra hyphen in the manuscript. Please do a thorough review).
Thank you for your comment. The hyphen has been checked along the manuscript.
Lines 66, 74, 90, 93, etc.. Remove the hyphen: Membrane.
It has been done. Thank you.
Line 108: Incomplete abbreviation for pvdf.
It has been added.
Line 109: PS instead of PUS?
Thank you. It has been modified.
Line 149. Superscript for m2. (Same comment at line 154).
It has been modified in both sentences.
Rephrase line 159.
The sentence has been rephrased as suggested by the reviewer.
Table 1. Table should be presented in table format. Not image.
As suggested by the reviewer the Table 1 is presented in table format
Table 1 b), Quality too low for reading.
It is true that the resolution was not good enough as indicated by the reviewer. In the revised version of the manuscript Table 1 is presented in table format, not as an image. Thus, its resolution is good now.
Lines 799-801. I am not sure about this sentence. At the line 72, it is mentioned that UF has the largest market share.
The sentence has been modified.

Reviewer 3 Report
The presented manuscript reviews some of the technological developments in the area of membrane filtration for dairy applications. It points out economic drivers for membrane filtration in dairy and suggests processing options for an increased integration of membrane filtration technology in processes to improve sustainability.
The manuscript presents a comprehensive overview over the subject. However, I hop that the comments and concerns I will detail below, will be carefully addressed to improve the quality of the manuscript, its focus, and readability.
General comments:
- The title says the manuscript deals with 'technology', but chapters 2 and 3 talk about economic aspects that are not relevant for the core of the manuscript which is the suggestion of possible processing routes for dairy applications. Please shorten these sections considerably. It will improve the focus and enhance the experience for the reader as a 25 pages review is very lengthy.
- The suggested 'traditional' processing steps that can be reconfigured or replaced by using membranes are not really an innovation that the authors can claim for themselves nor are these taken from most recent literature Every larger equipment manufacturer that offers membrane filtration units (GEA, TetraPak, Alfa Laval) does have brochures with comprehensive maps and flowcharts depicting membrane filtration unit operations in dairy processing (https://www.google.com/url?sa=t&rct=j&q=&esrc=s&source=web&cd=&ved=2ahUKEwiJy-6Gid7zAhVyk-AKHczGC2MQFnoECBoQAQ&url=https%3A%2F%2Fwww.gea.com%2Fen%2Fbinaries%2Fmembrane-filtration-in-dairy-industry_tcm11-17109.pdf&usg=AOvVaw1uMv_vi4u9vvutIqIDjNNQ)
Specific comments:
L21-24: Membrane filtration is state-of-the-art in the dairy industry. I think membrane filtration can be categorized in 1) replace existing processes, 2) complement existing processes to enhance yield, reduce energy or capital expenditure or increase sustainability by producing less waste, but more foo or feed.
L30: Progress is ongoing and so it does in membrane filtration. Membrane filtration has been around since the 1970s and i would not expect a big leap forward in the coming years, but gradual implementation.
L34-43 This is very general and can be summarized and shortened. Focus on your topic.
L47: 'Yellow line/products" will not be internationally recognized. This is a European dairy term.
L52-57: not relevant. Shorten or remove.
L74: Please remove all the dashes like 'mem-brane' in the text in all words where it might have appeared by accidentthrough copy-past of 'Ctrl+Shift+Dash'. I found at least 10.
L105-110: Some of your summaries are very generic and these loose accuracywhen membrane materials, pore sizes, MWCO, applications, membrane manufacturers, applications of membranes are not well separated and defined. gathering loose facts is something I observed throughout the manuscript. Please distinguish pressure driven membrane separation from ED and FO, note that certain polymers can only yield certain pore sizes and used for specific applications. Membrane filtration is pressure driven separation based on size. molecules have a size distribution and so do pores. There is no clear cut-off.
Table 1: Alfa Laval, one of the most important membrane manufacturers for the dairy industry in Europe is completely left out.
L153: Please be aware what interjections to connect sentences, such as 'Thus' 'Certainly' means. There are a few instances, where these logic connection do not make sense at all in the context.
L175-180: not clear how this are related to each other here?
Figure 1 a and b: I think this can be summed up in two sentences. N need to present graphs without a y-axis scaling.
General comment: You summarize or taking a lot of summaries from other summaries. This does not improve the review as this means you are not citing primary literature, but profiting from other's work/summaries.
L197: "among others"...please remove such unspecific interjections and be specific. The reader sees questions marks when reading this. It is great that you might know it, but the reader has a legitimate interest to know it too, if relevant.
L221-228: not relevant. Not well explained why relevant.
Figure 2a: If numbers are not normalized against the overall increase in publications in the field of dairy or membrane filtration, this has little value. The number of publications in every innovative field is increasing exponentially.
Table 2: Membrane applicability" is not clear as membranes can create permeate and retentate, fractionate and concentrate. So what can be achieved is not clear.
Figure 3: The most important'pasteurization' step is missing.
L305-340: This text is not relevant for your review. You should resist the temptation to write about dairy processing and everything you might know.
L345: 'some components'...be specific!
L417: 4-5 log(!)
L420: this refers to 'late blowing'...b specific!
L430-450: too generic introduction.
Figure 6: I think the order of processing steps and MF and UF are confused?
L474: I think it makes no sense to concentrate first by RO and then add water for diafiltration?
Figure 8: The text says you start with RO concentrate, not skim whey?
502: The most common term for NCPP is MCC today.
L525: More recent work by Toro-Sierra J and Haller, N could improve the efficiency of the process.
Figure 11: UF pore size 50 nm? The size of b-Lg and a-la is 3-4 nm?
L538-543: I think this is not relevant here in this context?
L587-588: Sentence not logical in this context.
table 3: Does this table really have content? I don't think so.
L736-737: Very generic and does not fit here as a single paragraph.
L782-786: Very generic.L798: GOS as a high-value product and purified by membranes remain completely unmentioned in your review.
L799-800: I think UF is still the most widely used membrane, at least by membrane area!
L823-833: This section is vague and not very specific. rewrite.
Author Response
The presented manuscript reviews some of the technological developments in the area of membrane filtration for dairy applications. It points out economic drivers for membrane filtration in dairy and suggests processing options for an increased integration of membrane filtration technology in processes to improve sustainability. The manuscript presents a comprehensive overview over the subject. However, I hope that the comments and concerns I will detail below, will be carefully addressed to improve the quality of the manuscript, its focus, and readability.
We acknowledge the time to revise the document as well as the comments provide.
General comments:
The title says the manuscript deals with 'technology', but chapters 2 and 3 talk about economic aspects that are not relevant for the core of the manuscript which is the suggestion of possible processing routes for dairy applications. Please shorten these sections considerably. It will improve the focus and enhance the experience for the reader as a 25 pages review is very lengthy.
Thank you for your comment. As suggested by another reviewer, the title has been modified to be more general and not only focused on the technology part. On the other hand, a table and some parts of the manuscript have been removed. Moreover, although the proposal of reducing sections 2 and 3 was facing a previous revision where it was suggested to add more information to have more quality as a review, two figures of section 3 have been removed and the text of sections 2 and 3 has been changed according to reviewer’s comments. Finally, all suggestions and comments provided by the reviewer have been addressed in this letter and have been considered in the preparation of the revised manuscript.
The suggested 'traditional' processing steps that can be reconfigured or replaced by using membranes are not really an innovation that the authors can claim for themselves nor are these taken from most recent literature Every larger equipment manufacturer that offers membrane filtration units (GEA, TetraPak, Alfa Laval) does have brochures with comprehensive maps and flowcharts depicting membrane filtration unit operations in dairy processing (https://www.gea.com/en/binaries/membrane-filtration-in-dairy-industry_tcm11-17109.pdf)
Throughout the document, the word “innovation” was limited when referring to already commercial processes since, as the reviewer says, companies such as GEA already have membrane technology in this field. Moreover, section 2 describes the already commercial membranes in dairy industries, such as membranes developed by GEA industries.
Specific comments:
L21-24: Membrane filtration is state-of-the-art in the dairy industry. I think membrane filtration can be categorized in 1) replace existing processes, 2) complement existing processes to enhance yield, reduce energy or capital expenditure or increase sustainability by producing less
waste, but more foo or feed.
Thank you for your remark. The text has been changed accordingly.
L30: Progress is ongoing and so it does in membrane filtration. Membrane filtration has been around since the 1970s and i would not expect a big leap forward in the coming years, but gradual implementation.
The sentence has been modified.
L34-43 This is very general and can be summarized and shortened. Focus on your topic.
It is true that the paragraph is very general, but it was the idea as it is the first paragraph of the introduction section. However, the paragraph has been rephrased as suggested by the reviewer.
L47: 'Yellow line/products" will not be internationally recognized. This is a European dairy term.
You are correct. It has been changed.
L52-57: not relevant. Shorten or remove.
As suggested by the reviewer the text has been removed.
L74: Please remove all the dashes like 'mem-brane' in the text in all words where it might have appeared by accident through copy-past of 'Ctrl+Shift+Dash'. I found at least 10.
Thank you for your remark. The words have been checked along the manuscript.
L105-110: Some of your summaries are very generic and these loose accuracy when membrane materials, pore sizes, MWCO, applications, membrane manufacturers, applications of membranes are not well separated and defined. gathering loose facts is something I observed throughout the manuscript. Please distinguish pressure driven membrane separation from ED and FO, note that certain polymers can only yield certain pore sizes and used for specific applications. Membrane filtration is pressure driven separation based on size. molecules have a size distribution and so do pores. There is no clear cut-off.
Following the suggestion of the reviewer an effort has been done to distinguish between pressure-driven membrane processes and membrane processes driven by other forces, such as electrical or chemical.
Table 1: Alfa Laval, one of the most important membrane manufacturers for the dairy industry in Europe is completely left out.
Sorry for the inconvenient. Information about company Alfa Laval has been incorporated into the revised manuscript (page 5, lines 449-451).
L153: Please be aware what interjections to connect sentences, such as 'Thus' 'Certainly' means. There are a few instances, where these logic connection do not make sense at all in the context.
An effort to avoid such non-appropriate interjections has been done.
L175-180: not clear how this are related to each other here?
As indicated by the reviewer the text has been modified to provide continuity between both sentences.
Figure 1 a and b: I think this can be summed up in two sentences. N need to present graphs without a y-axis scaling.
As indicated by the reviewer, both graphs have been removed and only few sentences are used to describe them.
General comment: You summarize or taking a lot of summaries from other summaries. This does not improve the review as this means you are not citing primary literature, but profiting from other's work/summaries.
As indicated by the reviewer and effort has been done in order to provide original contribution of the reviewed information. For that, a plagiarism test has also been done to corroborate that the manuscript accomplishes the requirements of the journal.
L197: "among others"...please remove such unspecific interjections and be specific. The reader sees questions marks when reading this. It is great that you might know it, but the reader has a legitimate interest to know it too, if relevant.
As indicated by the reviewer the use of this interjection has been removed and then the cited knowledge has been specified to avoid any question mark.
L221-228: not relevant. Not well explained why relevant.
As indicated by the reviewer, the paragraph was not relevant. The text has been re-structured to address the lack of relevance stated in the sentence and used it more properly.
Figure 2a: If numbers are not normalized against the overall increase in publications in the field of dairy or membrane filtration, this has little value. The number of publications in every innovative field is increasing exponentially.
As suggested by the reviewer the number of publications of Figure 2a (Figure 1a in the new version of the manuscript) has been normalized.
Table 2: Membrane applicability" is not clear as membranes can create permeate and retentate, fractionate and concentrate. So what can be achieved is not clear.
The last column of Table 2 has been removed in order to do not confuse the reader.
Figure 3: The most important 'pasteurization' step is missing.
Figure 3 (Figure 2 in the new version of the manuscript) is a simplified scheme of the process, where pasteurization and UHT are depicted as a black box.
L305-340: This text is not relevant for your review. You should resist the temptation to write about dairy processing and everything you might know.
The text has been modified and slightly reduced.
L345: 'some components'...be specific!
As indicated, the sentence has been modified.
L417: 4-5 log(!)
The text has been modified to clarify that was referring to the reduction factor of bacteria and spores.
L420: this refers to 'late blowing'...b specific!
As indicated the text has been modified to clarify the meaning of the sentence.
L430-450: too generic introduction.
As indicated, generic concepts have been removed from this section.
Figure 6: I think the order of processing steps and MF and UF are confused?
As suggested, Figure 6 (Figure 5 in the new version of the manuscript) has been modified.
L474: I think it makes no sense to concentrate first by RO and then add water for diafiltration?
RO is used to remove undesired components and concentrate protein and thus, purifying the whey (Figure 6 of the new version of the manuscript). After RO, the purified skim whey is introduced into an UF system in order to produce WPC. Then, WPI can be obtained by a diafiltration step, after the UF (Figure 7 of the new version of the manuscript). In order to clarify this point, the text has been changed accordingly.
Figure 8: The text says you start with RO concentrate, not skim whey?
As shown in Figure 6 of the new version of the manuscript, the RO retentate is composed by skim whey. Therefore, Figure 8 (Figure 7 in the new version of the manuscript) starts skim whey.
502: The most common term for NCPP is MCC today.
MCC abbreviation has been introduced in the revised version of the manuscript.
L525: More recent work by Toro-Sierra J and Haller, N could improve the efficiency of the process.
As suggested the recent work of Toro-Sierra J and Haller has been reviewed and their contribution introduced in the text (page 8, lines 582-591).
Figure 11: UF pore size 50 nm? The size of b-Lg and a-la is 3-4 nm?
During the last years, UF pore sizes have been reduced up to 40-50 nm, especially when trying to provide high removal ratios of virus. Although L-g and a-La sizes approaches to 3-4 nm, rejection mechanisms of UF membranes are not only depended on solute size and pore size but also on molecule geometry and structure, since they are relevant in rejection values.
L538-543: I think this is not relevant here in this context?
As suggested the paragraph has been removed.
L587-588: Sentence not logical in this context.
The sentence has been removed.
Table 3: Does this table really have content? I don't think so.
Table 3 has been removed.
L736-737: Very generic and does not fit here as a single paragraph.
The sentence has been removed.
L782-786: Very generic.
As suggested the text has been modified.
L798: GOS as a high-value product and purified by membranes remain completely unmentioned in your review.
As suggested the relevance of GOS as a high-value product and its purification by membranes has been introduced in the text (see page 10, lines 634-645).
L799-800: I think UF is still the most widely used membrane, at least by membrane area!
UF has the largest market share and it is also widely used. The text has been modified accordingly.
L823-833: This section is vague and not very specific. rewrite.
As suggested the text of this section has been modified.

Reviewer 4 Report
The paper entitled "Technology perspectives on by-products from dairy industry by using membrane technologies integration in conventional treatments" is a well-structured and written paper in the field of membrane technologies towards the processing of dairy products. The authors are specialists in such relevant applications of membranes. Their overview, findings and discussion are good. From my point of view, the paper needs to attend to some minor comments:
1) Please, give feedback regarding the economical aspects of membranes in the dairy industry.
2) Concluding remarks: As experts in the field, the authors can also state in a clear manner the future trends and perspectives in the field, along with the recommendations for the new researchers in the field.
Author Response
The paper entitled "Technology perspectives on byproducts from dairy industry by using membrane technologies integration in conventional treatments" is a well-structured and written paper in the field of membrane technologies towards the processing of dairy products. The authors are specialists in such relevant applications of membranes. Their overview, findings and discussion are good.
We acknowledge the positive judgment of the reviewer.
From my point of view, the paper needs to attend to some minor comments:
1) Please, give feedback regarding the economical aspects of membranes in the dairy industry.
The efforts on the economic aspects of membranes in the dairy industry have been discussed in section 3 (Market role of membrane technologies on dairy industry). A large effort on collection of economic data or market value of dairy industry were addressed to open information that was summarized in section 3. More detailed information is collected in economic reports in non-open mode for which the cost of each of them was not in our funding capacities.
2) Concluding remarks: As experts in the field, the authors can also state in a clear manner the future trends and perspectives in the field, along with the recommendations for the new researchers in the field.
As suggested by the reviewer, the conclusions section has been re-written to stress which could be the expected successful integration of new membrane technologies. Indeed, they will be those centered in: i) new NF membranes with new active layer surface chemistries to exploit the separation factors or promote a specific removal of un-desired ionic components; ii) the integration of ED and ED with bipolar membranes to promote added value by-products recovery, taking into account the development of more mono-selective membranes and bipolar membranes by companies as Amstom. Suez or FujiFilm; and iii) the promising development of FO membranes as those completed by Aquaporin company. However, in some cases, such as in the last case, there are not still industrial scale applications reported by this company and all the development is still at piloting stage, according to personnel communication. The progress on the state of the art, was recommended to be removed from the text in a previous review stage to focus the review on the existing membrane technology processes.

Round 2
Reviewer 3 Report
Thank you for considering all the comments of the reviewers. The manuscript is now acceptable for publication.